# Gel Property of Soy Protein Emulsion Gel: Impact of Combined Microwave Pretreatment and Covalent Binding of Polyphenols by Alkaline Method

**DOI:** 10.3390/molecules27113458

**Published:** 2022-05-27

**Authors:** Jing Xu, Fei Teng, Baiqi Wang, Xinxuan Ruan, Yifan Ma, Dingyuan Zhang, Yan Zhang, Zhijun Fan, Hua Jin

**Affiliations:** 1College of Arts and Sciences, Northeast Agricultural University, Harbin 150030, China; xujing@neau.edu.cn (J.X.); tengfei19980910@163.com (F.T.); 18845043529@163.com (B.W.); ruanxinxuan@163.com (X.R.); m1194665331@163.com (Y.M.); zdy4507@163.com (D.Z.); 2Coastal Research and Extension Center, Mississippi State University, Starkville, MS 39762, USA; yzhang@fsnhp.msstate.edu; 3Heilongjiang Beidahuang Green and Healthy Food Co., Ltd., Jiamusi 154007, China; 15845177666@139.com

**Keywords:** microwave modification, alkali polyphenol covalently combined modification, combined modification, emulsion gels, gel properties

## Abstract

This study investigated the effects of microwave modification, alkali polyphenol (ferulic acid) covalently combined modification, and microwave-alkali polyphenol covalently combined modification on the gel properties of soy protein emulsions. The results showed that the properties of soy protein emulsions were improved significantly by the three modification methods. After three kinds of modification, the viscoelasticity of soy protein emulsion gel increased, and a gel system with stronger elasticity was formed. The texture, water-holding, and hydration properties of the emulsion gel increased significantly. The SEM and ClSM results showed that the modified soy protein emulsion gel had a more compact and uniform porous structure, and the oil droplets could be better embedded in the network structure of the gel. Among the three modification methods, the microwave-alkali method polyphenol covalently combining the compound modification effect was best, and the microwave modification effect was least effective compared to the other two methods. Our obtained results suggested that for gel property modification of soy protein emulsion gels, microwave pretreatment combined with the covalent binding of polyphenols by an alkaline method is an effective method.

## 1. Introduction

Protein emulsion gel is a kind of gel that is formed by adding the oil phase to a protein solution to form a stable emulsion with protein as an emulsifier and then gelling the emulsion to cross-link the protein [1,2,3]. Compared with protein gels, protein emulsion gels are easier to prepare. The oil droplets in the three-dimensional network structure of the emulsion gel can be used as active fillers to support the three-dimensional structure of the gel. The emulsion gel has better stability and rheological properties [1]. At present, protein emulsion gel has been widely used in many foods, such as yogurt and sausage [4]. At the same time, the oil droplets in the protein emulsion gel and the structure of the gel network bind the water molecules so that protein emulsion gel can be used as the carrier system of hydrophobic and hydrophilic active substances. In recent years, protein emulsion gels have received extensive attention as carriers of carotenoids, vitamins, unsaturated fat, etc. [5].

Soybean protein isolate (SPI) is an ideal substitute for animal protein due to its rich nutritional value and excellent functional properties, such as emulsifying properties, gelation properties, solubility, and foaming properties [6,7]. The emulsion gel prepared from SPI can reduce the content of animal protein and fat in the food system, which is suitable for low-fat food systems [8]. In addition, SPI is made mainly from defatted soybean meal, a byproduct of oil extraction. Therefore, the preparation of soy protein emulsion gel can effectively improve the utilization ratio and economic added value of soy products [9]. However, natural protein cannot fully meet the needs of the food processing system. Research shows that the functional properties of proteins can be improved by various modification techniques, and special protein products with better functional properties can be obtained [6].

Microwave modification can make the side chain groups of proteins obtain energy to produce active free radicals in the microwave field, which results in the enhancement of the interaction within and between protein molecules and the change in protein structure, improving the functional properties of proteins [10]. The effect of microwave treatment on the functional properties of rice bran protein was studied by Khan et al. [11]. The results showed that microwave treatment could expose more hydrophobic groups in the protein molecules, promote the protein to have a better hydrophilic–hydrophobic balance, and improve the emulsifying and foaming properties of the protein. Protein–polyphenol basic complexes were obtained by covalent binding of the protein and polyphenol under alkaline oxygen conditions, and C-S bonds or C-N bonds could be formed between them [12]. Sui [13] showed that the secondary structure of soybean protein-anthocyanin covalent complexes changed after soybean protein and anthocyanin covalent binding under alkaline conditions, and their emulsifying and foaming properties were better than those of natural soybean proteins. The results showed that the gel properties of protein gels prepared by modified proteins were improved significantly. Zhao [14] used ultrasonic treatment to modify soybean protein. The results showed that the storage/elastic modulus, gel strength, and water-holding capacity of the soybean protein cold gel were improved. The microstructure of the gel was denser and more homogeneous, and the pores were smaller. Guo [15] showed that the interaction between soybean protein and tannic acid improved the structure and rheological properties of soybean protein gels. Different modification methods are also used in the preparation of protein emulsion gels. Whey protein emulsion gels formed by the acid-induced modification of whey protein have a higher storage modulus and breaking force [4]. The modified whey protein was prepared by Lu [16], who showed that the denaturation of the modified whey protein could improve the mechanical properties of the gel, such as fracture stress, Young’s modulus, and the storage modulus. Qayum et al. [17] modified α-lactalbumin by ultrasound pretreatment and laccase cross-linking. The microstructure of the gel was more homogeneous and compact and was stored for more than 1 month without any separation.

At present, there are few studies on improving the gel properties of protein emulsions by microwave modification or polyphenol covalent modification. There are few reports on the preparation of protein emulsion gels with good properties by compound modification of proteins. Therefore, in this study, microwave modification, alkaline polyphenol covalent modification, and microwave-alkaline polyphenol covalent modification were used to modify soybean protein. The effects of different modification methods on the properties of soy protein emulsions (mean particle size, zeta potential, interfacial protein content, apparent viscosity) and gel properties (rheological analysis, texture properties, water-holding capacity, hydration properties, microstructure) were studied, and the action mechanism and application effect of modified soy protein as an emulsifier and gel matrix were investigated to provide a theoretical basis for the development of a high-performance soy protein emulsion gel system.

## 2. Results and Discussion

### 2.1. Properties of Soy Protein Emulsion

#### 2.1.1. Mean Particle Size, Zeta-Potential of Emulsion

The average particle size of the emulsion can reflect the stability of the emulsion. The smaller the average particle size of the emulsion, the more stable the emulsion [18]. The average particle sizes of different modified soy protein emulsions are shown in Table 1. After microwave treatment, the average particle size of the MSPI emulsion decreased significantly (*p* < 0.05), attributed to the expansion of the molecular structure of the protein by microwave treatment, which makes it better adsorbed on the oil–water interface to form an emulsion with a smaller average particle size [19]. After covalent binding with polyphenol, the average particle size of the SPI-FA (AM) emulsion decreased from 282.77 ± 0.67 nm to 243.07 ± 0.25 nm, possibly because the emulsifying property of soy protein emulsions increases after covalent binding with polyphenols, which leads to a decrease in the average particle size and an increase in the stability of soy protein emulsions [20]. After the soy protein was covalently modified by microwave-alkaline polyphenols, the average particle size of the emulsion decreased further. Microwave modification and alkaline polyphenol covalent modification can possibly promote each other to improve the stability of soy protein emulsions and form emulsions with a smaller average particle size and more stable performance. Wang et al. [21] showed that the covalent modification of EGCG and glycosylated black soybean protein could further reduce the average particle size of black soybean protein emulsions. The results of this study are similar to the average particle size of soy protein emulsions modified by this method.

The higher the absolute value of the zeta potential, the greater the electrostatic repulsion between emulsion droplets, and the smaller the aggregation of emulsion droplets caused by the interaction, the more stable the emulsion [22]. The zeta potentials of different modified soy protein emulsions are shown in Table 1. After microwave treatment, the zeta potential of the soybean protein emulsion increased from 26.20 ± 0.30 mV to 28.17 ± 0.42 mV, possibly due to the ability of microwave treatment to unfold the structure of protein, exposing more charged groups, resulting in an increase in the surface charge of protein molecules and the formation of an emulsion with a greater zeta-potential absolute value [23]. After covalent binding with polyphenol, the zeta potential of the soybean protein emulsion increased from 26.20 ± 0.30 mV to 29.60 ± 0.10 mV. The results showed that the addition of polyphenols increased the electrostatic repulsion between the emulsion droplets and improved the stability of the emulsion, which is consistent with the results of Zhao et al. [24]. After soy protein was covalently modified by microwave-alkaline polyphenols, the absolute value of the zeta potential of the emulsion increased further based on a single modification method.

#### 2.1.2. Interfacial Protein Adsorption of Emulsion

The interfacial protein adsorption (AP) reflects the percentage of protein at the oil–water interface. Increasing AP% can effectively increase the interface thickness of the emulsion, which is beneficial to the preparation of the emulsion with better physical stability [25]. The interfacial protein adsorption (AP) of soy protein emulsions modified in different ways is shown in Table 1. After microwave treatment, the interfacial protein adsorption of the MSPI emulsion increased significantly (*p* < 0.05), possibly because the structure of soy protein is unfolded by microwave modification, which exposes more hydrophobic groups and makes the soy protein have a better hydrophobic balance, which results in more protein molecules adsorbing on the oil–water interface, increasing the interfacial protein adsorption [21]. After covalent binding with ferulic acid (FA), the interfacial protein adsorption of the SPI-FA (AM) emulsion increased from 40.27 ± 0.65% to 69.67 ± 2.52%, which can be attributed to the introduction of polyphenol changing the molecular conformation of proteins, making more protein molecules adsorb on the surface of oil drops more quickly, increasing the protein content at the emulsion interface, increasing the thickness of the oil–water interface film in the emulsion, and enhancing emulsion stability [12,21,22,23,24,25,26]. The results are consistent with the average particle size of the soy protein emulsion in Table 1. In addition, studies by Sabouri [27] et al. have shown that polyphenols can reduce the surface tension of protein molecules and promote their adsorption at the oil–water interface. After microwave-alkaline polyphenol covalent modification, the interfacial protein adsorption of the MSPI-FA(AM) emulsion increased further, showing that the compound modification method is more effective than the single modification method in improving the adsorption of protein on the surface of oil drops and more effective in increasing the interfacial protein adsorption, which is helpful to prepare a more stable protein emulsion system.

#### 2.1.3. Apparent Viscosity of the Nanoemulsion

The emulsion stabilized by soy protein is a non-Newtonian fluid emulsion with shear thinning behavior, and the apparent viscosity decreases with increasing shear rate [28]. The apparent viscosity curves of different modified soy protein emulsions were analyzed by the Ostwald–de Waele empirical formula, and the limiting high-shear viscosity (η_∞_), the consistency coefficient (K), and flow characteristic index (*n*) were fitted as shown in Table 2. In general, the greater the K value, the greater the viscosity of the emulsion. The closer the n value is to 0, the more obvious the shear thinning. The closer the n value is to 1, the lower the dependence of the emulsion viscosity on the shear rate [13]. After modification by microwaves, the viscosity of the emulsion decreased, and the n value increased from 0.616 ± 0.010 to 0.754 ± 0.004, which showed that the dependence of the emulsion viscosity on the shear rate decreased, possibly due to the change in the conformation of soy protein modified by microwaves, the formation of emulsions with higher zeta-potential absolute values, the decrease in aggregation between emulsion droplets, and the decrease in shear thinning behavior of emulsions [29]. After covalent binding with polyphenol, the viscosity coefficient K of the SPI-FA(AM) emulsion decreased, and the n value increased from 0.616 ± 0.010 to 0.768 ± 0.004, which showed that covalent binding with polyphenol could decrease the viscosity of the emulsion stabilized by protein. The apparent viscosity was reduced by the shear rate, which can be attributed to the decrease in the average particle size of the emulsion stabilized by the polyphenol-protein covalent complex, the decrease in the flow resistance, the greater order under shear conditions, the decrease in the viscosity of the emulsion, and the lower shear-thinning behavior of the emulsion [30]. After the soy protein was covalently modified by microwave-alkaline polyphenols, the viscosity of the emulsion was further reduced, and the n value was higher, which indicated that droplet aggregation was less and that the shear thinning behavior was weaker (Figure 1).

### 2.2. Rheological Analysis of Soy Protein Emulsion Gel

The gelation dynamics and viscoelasticity of emulsion gel are reflected mainly by the change in elastic modulus (G′) and viscosity modulus (G″) in the small amplitude oscillatory rheological test. G′ represents the elastic property of the gel, and G″ represents the viscous property of the gel. The larger G′ is, the more the gel resembles a solid material, and the larger G″ is, the more the gel resembles a liquid material [31]. The time-sweep tests of soy protein emulsion gels modified in different ways are shown in Figure 2A. The crossover points between G′ and G″ of different emulsion gels are shown in Table 3. t_c_ is the abscissa of the crossover point and G_c_ is the ordinate of the crossover point. The dependence of the G′ and G″ on angular frequencies (ω, rad/s) can be characterized by frequency scanning of the emulsion gel, as shown in Figure 2B. As shown in Figure 2A, with increasing time, G′ and G′’ increase, and the rate of increase of the G′ value is greater than the rate of increase of the G″ and finally reaches G′ > G″, which indicates the formation of an emulsion gel with solid properties, because GDL can shield the surface charge of soy protein so that the electrostatic repulsion between protein molecules is reduced, which results in the cross-linking of soy protein molecules and the formation of the network structure of the emulsion gel [32]. As time goes on, G′ and G′’ tend to be stable, and the larger the final G′ value is, the stronger the structure of the emulsion gel is. As shown in Figure 2A, the final G′ value of the soy protein emulsion gel after microwave treatment was higher than the final G′ value of the natural soy protein emulsion gel, which shows that microwave treatment can enhance the viscoelasticity of the emulsion gel and accelerate the formation of gel. The results are similar to the results reported by Qin et al. [33]. After covalent binding with polyphenols, the final G′ value of the SPI-FA (AM) emulsion gel was higher than that of the natural soy protein emulsion gel. The gel formed too fast for the machine to catch. This may be due to the covalent binding of polyphenol to form an emulsion with a smaller average particle size, which results in the formation of more cross-linking sites per unit volume and the enhancement of the network structure of the emulsion gel [34]. The final G′ value of the MSPI-FA (AM) emulsion gel was higher after microwave-alkaline polyphenol covalent modification, possibly because the structure of soy protein in the SPI-FA (AM) complex is further expanded by microwave treatment, the active groups inside the molecule are exposed, and stronger cross-links are formed during the gelation process, which leads to an increase in the strength of the gel network [13]. The same results can be observed in Figure 2B. The G′ and G′’ of the modified soy protein emulsion gel are larger than the G′ and G′’ of the natural soy protein emulsion gel, respectively, which indicates that the modified soy protein emulsion gel has a stronger network structure.

The parameter power law constant (K′) and frequency index (n′) can be obtained by fitting the G′-ω curve with the power law type model [35], as shown in Table 4. The value of n′ is an important parameter to reflect the viscoelasticity of emulsion gel, and n′ = 0 is pure elastic gel. The larger n′ is, the weaker the elastic properties are in the gel structure [36]. As seen from Table 4, the n′ value of the natural soy protein emulsion gel is close to 0.1, which indicates that the natural soy protein emulsion gel belongs to the category of weak gel [37]. After microwave treatment, the value of K′ increased by approximately 36.4%, and the value of n′ decreased by 9.5%, which showed that microwave treatment could enhance the elastic properties of soy protein emulsion gel so it was a more elastic solid material, which is consistent with the results of Qin et al. [33]. After covalent binding with polyphenol, the value of K′ increased significantly (*p* < 0.05), and the value of n′ decreased significantly (*p* < 0.05). These results showed that the covalent modification of polyphenols could improve the gel properties of soy protein emulsion gels, which is similar to the results of Guo [15]. After microwave-alkali-polyphenol covalent modification, K′ further increased and n′ further decreased. The results showed that compared with the single modification method, the microwave-alkaline polyphenol covalent modification could improve the properties of soy protein emulsion gel to a great extent, and the emulsion gel had stronger elastic properties.

### 2.3. Gel Properties of Soy Protein Emulsion

#### 2.3.1. Textural Properties of Soybean Protein Emulsion Gel

The textural properties (hardness, elasticity, cohesiveness, chewiness) of the emulsion gel are shown in Table 5. After microwave treatment, the hardness, elasticity, cohesiveness, and chewiness of the MSPI emulsion gel increased significantly (*p* < 0.05), possibly because microwave treatment enables soy protein to form emulsions with smaller average particle sizes, while smaller emulsion droplets cause more binding sites to form per volume during gel formation, which makes the internal structure of the emulsion gel more compact and uniform and improves the texture properties of the emulsion gel [38]. The hardness, elasticity, cohesiveness, and chewiness of the modified soy protein emulsion gel increased significantly (*p* < 0.05) because, under alkaline conditions, ferulic acid is oxidized to form semiquinones, which can covalently bind to proteins. After covalent binding, nonpolar groups are introduced into the protein molecules, which leads to the enhancement of hydrophobic interactions in the emulsion gel, the stability of the network structure of the gel, and the enhancement of the texture properties of the emulsion gel [12]. After microwave treatment with SPI-FA (AM), the texture properties of the emulsion gel were further improved, due mainly to the ability of microwave treatment to unfold the structure of proteins in the complex molecules, exposing hydrophobic groups, hydrophilic groups, and free sulfhydryl groups to the surface of the complex molecules [39]. During gel formation, these exposed groups formed stronger intermolecular forces, which supported the three-dimensional network structure of the emulsion gel and resulted in better texture properties of the emulsion gel.

#### 2.3.2. Water-Holding Capacity of Emulsion Gel

The water-holding capacity of the emulsion gel network reflects the ability to maintain water molecules and the structural stability of the emulsion gel [1]. After different modification methods, the water-holding capacity of soy protein emulsion gel is shown in Table 5, which shows that the water-holding capacity of the emulsion gel increased significantly after microwave modification (*p* < 0.05) because the average particle size of soy protein emulsion gel modified by microwaves is smaller, and these emulsion droplets form more cross-linking units during the formation of emulsion gel, which can lead to a denser and more homogeneous structure of emulsion gel network [40]. The principle of the water-holding capacity of the emulsion gel is that the porous structure of the gel can form capillaries, which can cause capillarity to the water molecules and then adsorb them in the three-dimensional network structure of the gel [41]. The water-holding capacity of the emulsion gel is increased because of the uniform and denser structure of the gel network caused by microwave modification. Table 5 also shows that the covalent binding of polyphenols resulted in a significant increase in water-holding capacity (*p* < 0.05) due to the enhancement of the structure of the gel network by the intervention of polyphenol and the formation of a more compact and uniform porous network by the cross-linking of small emulsion droplets, which leads to the enhancement of the water-holding capacity of the gel [42]. In addition, the water-holding capacity of the SPI-FA (AM) emulsion gel was further enhanced after microwave treatment. These results showed that polyphenol modification and microwave modification could affect the water-holding capacity of the emulsion gel, and they played a synergistic role.

#### 2.3.3. Hydration Characteristics of Emulsion Gel

The water molecules in the gel material can behave in an easy-to-flow state and free-flow state. The law of the exchange and transfer of H^1^ protons in the gel water molecules of the emulsion can be obtained by the low-field nuclear magnetic resonance technique; furthermore, the hydration properties of the emulsion gel were characterized [19]. The spin–spin relaxation time (T_2_) including T_2b_, T_21_, T_22_, and T_23_ is the time corresponding to each peak in the relaxation diagram. The shorter the peak time (that is, the smaller T_2_), the more difficult it is for water molecules to flow [43]. In general, T_2B_ (<1 ms) represents strong-binding water that is electrostatic and hydrogen-bonded to protein molecules, T_21_ (1~10 ms) represents weak-binding water that is tightly bound to protein molecules, T_22_ (10~100 ms) denotes water that does not easily flow in the pores of the gel network, and T_23_ (>100 ms) denotes free-flowing water outside the gel network structure [44]. P_2B_, P_21_, P_22_, and P_23_ represent the percentage of peak area (%) under T_2b_, T_21_, T_22_, and T_23_, respectively, which shows the percentage of strong-binding water, weak-binding water, hard-flowing water, and free-flowing water. T_2B_ and T_21_ are not discussed in this text, because the binding water is not normally affected. Table 6 shows T_22_, T_23_, P_22_, and P_23_ of soy protein emulsion gels. The results showed that the relaxation time T_22_ and T_23_ decreased (*p* < 0.05), the content of flowing water increased (*p* < 0.05), and the content of free-flowing water decreased (*p* < 0.05), which can be attributed to the expansion of the structure of the soy protein modified by microwaves, the decrease in the average particle size of the emulsion, the formation of more cross-linking units per unit volume, the enhancement of the structure of the emulsion gel, the increase in the content of the difficult flowing water in the gel, and the free-flowing water content decline [45]. The relaxation time T_22_ and T_23_ decreased significantly (*p* < 0.05), the content of P_22_ increased significantly (*p* < 0.05), the content of P_23_ decreased significantly (*p* < 0.05), and the content of flowing water did not easily increase in the structure of the emulsion gel, possibly because proteins covalently bind with polyphenols to form a network of denser, uniform emulsion gels, resulting in more water molecules trapped in them in the form of less mobile water [46]. After covalent modification with microwave-alkali-phenol, T_22_ and T_23_ decreased further (*p* < 0.05), P_22_ increased further (*p* < 0.05), and P_23_ decreased further (*p* < 0.05). The results show that microwave-alkali-phenol covalent modification can enhance the binding ability of water molecules and increase the content of water in the porous structure of the gel. The results showed that microwave modification and the covalent modification of alkaline polyphenols had synergistic effects. The experimental results are consistent with the water-holding capacity of the emulsion gel (Figure 3).

### 2.4. Microstructure of Soy Protein Emulsion Gel

The microstructure of the soy protein emulsion gel was observed by scanning electron microscopy (SEM) and confocal laser scanning microscopy (CLSM), as shown in Figure 4. The SEM results showed that the oil droplets in the natural SPI emulsion gel could not be well embedded in the gel network, resulting in a loose and uneven network structure of the emulsion gel, possibly due to the large average particle size of natural SPI stabilized emulsions. Therefore, in the process of GDL-induced emulsion gel formation, the size of cross-linking units formed between emulsion droplets and proteins in the aqueous phase was different, which leads to the inhomogeneity of the network structure [47]. From the CLSM results, the larger oil droplets in the natural soy protein emulsion gel are seen to be distributed unevenly in the gel network. After modification in different ways, the network structure of the soy protein emulsion gel became denser and more uniform. Qin et al. [33] studied the effect of microwave treatment on the microstructure of soybean protein and wheat protein mixed gel. In the gelation process, the number of cross-linking sites per unit volume was increased, and the microstructure of the gel network was denser and more homogeneous. Balange et al. [48] studied the effect of covalent binding of alkaline polyphenols on Surimi gel. The results were consistent with the results of the present study. The microstructure of the soy protein emulsion gel was denser and more homogeneous after microwave-alkaline polyphenol covalent modification. The microstructure of different modified soy protein emulsion gels supported the previous experimental results; that is, the three-dimensional network structure of modified soy protein emulsion gels was denser and more homogeneous, and the texture, water-holding, rheological, and hydration properties of the emulsion gel were better (Figure 5).

## 3. Methods and Materials

### 3.1. Materials

Defatted soybean meal was purchased from Shandong Zhaoyuan Food Company Limited (Yantai, China). Ferulic acid was purchased from Shanghai Aladdin Biochemical Technology Company Limited (Shanghai, China). Soybean oil was purchased in Jiusan Grain and Oil Industry Group Company Limited (Harbin, China). All other chemical medicines were of analytical grade.

### 3.2. Extraction of Soybean Protein Isolate

According to the method of Jin [49], the defatted soybean meal was ground and sieved through 50 mesh to obtain the defatted soybean meal. Soybean protein isolate (SPI) was extracted from defatted soybean powder. Figure 6 shows the specific extraction process for SPI.

### 3.3. Preparation of Soybean Protein Isolate-Ferulic Acid Alkaline Conjugation

The soy protein-ferulic acid (FA) alkaline complex was prepared by the covalent binding of alkaline-method (AM) polyphenols according to the method of Yang et al. [50] with slight modification. The SPI lyophilized powder was dissolved in phosphate-buffered saline solution (PBS) (0.02 mol/L, pH 7.0), and FA was added to a final concentration of 2% in SPI (*w*/*v*) and a final concentration of FA of 150 mol/(g protein) (selected by our previous work). The solution was maintained at pH 9.0 with 2 mol/L NaOH and stirred for 12 h at room temperature. The reaction was then terminated at pH 7.0 with 2 mol/L HCl. The reaction solution was placed in an 8–14 kDa dialysis bag and dialyzed with PBS solution for 24 h. The resulting soy protein-ferulic acid alkaline complex was lyophilized and represented as SPI-FA (AM).

### 3.4. Microwave Treatment of SPI and SPI-FA(AM)

According to the method of Guan et al. [51], the sample was treated by microwave irradiation. SPI and SPI-FA(AM) freeze-dried powder were dissolved in 0.02 mol/L PBS (10%, *w*/*v*) at pH 7.0. After magnetic stirring for 2 h at room temperature, the samples were put into a 4 °C refrigerator to hydrate overnight. The sample solution was placed in a round-bottomed flask and microwave-treated using a microwave catalytic synthesis/extraction apparatus (XH-100A, Xianghu, Beijing, China) with a microwave power of 600 W and a microwave time of 4 min. Microwaves were selected by our previous work at 40 °C. Microwave-modified soy protein (MSPI) and microwave-alkaline acid polyphenol covalently combined modified soy protein (MSPI-FA (AM)) were prepared.

### 3.5. Preparation of Soybean Protein Emulsion

Based on the method of Feng et al. [25,52], a soybean protein emulsion was prepared by an ultrasonic method with natural SPI, MSPI, SPI-FA (AM), and MSPI-FA (AM) as emulsifiers. A 10 mL protein sample solution (protein concentration 10% (*w*/*v*)) and 1.76 mL of soybean oil were magnetically stirred for 10 min. The crude emulsion was then homogenized in a high-speed homogenizer (FJ200-SH, Sample Model Factory, Shanghai, China) (10,000 r/min, 2.5 min). The O/W emulsion was prepared by ultrasonic treatment (400 W, 15 min) with a titanium probe 0.636 cm in diameter in an ultrasonic cell grinder (JY92–2D, Scientz, Ningbo, China).

### 3.6. Preparation of Soy Protein Emulsion Gel

Based on the method of Li et al. [53], soy protein emulsion gel was prepared by adding gluconolactone (GDL) to soybean protein emulsion. The newly prepared soy protein emulsion was centrifuged at low speed (1000 r/min, 1 min) to remove the bubbles in the emulsion. Adding a certain amount of GDL powder to the emulsion, controlling the amount of GDL, the pH value of the emulsion gradually decreased and finally stabilized at 4.5. The soy protein emulsion gel was prepared by stirring the emulsion with GDL for 30 s at room temperature and then reacting at 40 °C for 2 h.

### 3.7. Characterization of Emulsions

#### 3.7.1. Average Particle Size, Zeta-Potential, and Polydispersity Index (PDI) Measurements

The average particle size and PDI of emulsion samples was determined by a Malvern Nano-S90 laser particle size analyzer (Malvern Nano-S90, Malvern Instruments, Worcestershire, UK) at 25 °C. The zeta potential of emulsion samples was measured by a zeta-sizer Nano zeta potential analyzer (zeta-sizer Nano Z, Malvern Instruments). To avoid multiple light scattering effects, the prepared emulsion was diluted 100 times with PBS solution (0.02 mol/L, pH 7.0) before measurement [54].

#### 3.7.2. Interfacial Protein Adsorption

According to the methods of Chen et al. [55]. and Nicole [56], the interfacial protein adsorption of emulsion samples was determined. The emulsion was centrifuged in a high-speed centrifuge (GL-20G-Ⅱ, Anting, Shanghai, China) (10000 r/min, 45 min). The top oil layer and the bottom protein layer were separated by the emulsion. The bottom solution was absorbed by a syringe and filtered through a 0.45 m filter membrane. To determine the concentration of protein in the filtrate, the formula for calculating the interfacial protein adsorption (AP) is as follows:(1)$$\mathrm{AP}(\%)=\frac{c_{s}-c_{f}}{c_{0}}\times 100\%$$
where $c_{f}$ is the protein concentration of the filtered subnatant from the emulsion; $c_{s}$ is the protein concentration of centrifugal supernatant from protein dispersion; and $c_{0}$ is the protein concentration of the initial protein dispersion.

#### 3.7.3. Apparent Viscosity

Based on the method of Xi et al. [57], the apparent viscosity of emulsion samples was determined by a rotational rheometer (Haake Mars 40, Thermo Fisher Scientific, Waltham, MA, USA). The new emulsion was placed between two parallel plates (radius 40 mm, distance 1 mm) and balanced for 30 s. The shear rate was 0.01–100 s^−1,^ and the temperature was 25 °C. The apparent viscosity (η) and shear rate (γ) can be analyzed by a power-law model known as the Ostwald–de Waele empirical formula:(2)$$\eta =K\times \gamma^{n-1}$$
where K is the consistency coefficient, γ is the shear coefficient, and n is the flow index.

### 3.8. Rheological Analysis of Soy Protein Emulsion Gel

According to Lopes-Da-Silva et al. [36], rheological analysis of soy protein emulsion gel samples was carried out by using a rotational rheometer (Haake Mars 40, Thermo Fisher Scientific, Waltham, MA, USA), fitted with a serrated plate-plate geometry (diameter 40 mm, gap 1 mm). After transferring the sample to the rheometer plate, the exposed sample surface was covered with a thin layer of mineral oil to avoid evaporation during the measurements.

Strain sweep: The linear viscoelastic region of the emulsion gel was obtained by placing the emulsion on the rheometer with a scanning strain of 0.001–100% and angular frequency of 1 rad/s at 40 °C. The results showed that the shear modulus of all samples did not obviously change when the strain was 0.003% and the angular frequency was 1 rad/s.

Time sweep: After adding GDL and magnetic stirring at 25 °C for 30 s, the emulsion was immediately placed between two parallel plates (radius 40 mm, distance 1 mm) and covered with silicone oil to prevent water dispersion. The scanning strain was 0.003%, the angular frequency was 1 rad/s, the temperature was 40 °C, and the time was 2 h. The storage modulus (G′) and loss modulus (G″) of the emulsion gel were obtained. G′ denotes the elastic property of the emulsion gel, and G″ denotes the viscous property of the emulsion gel. When G′ > G″, the sample was transformed from a liquid emulsion into a solid emulsion gel.

Frequency sweep: After 2 h of scanning, the emulsion gel was scanned by frequency scanning. The scanning strain was 0.003%, the angular frequency was 0.1–100 rad/s, and the temperature was 40 °C. The relationship between the storage modulus G′ and the angular frequency ω was fitted by the following formula:(3)$$G^{\prime }=K^{\prime }\times \omega^{n^{\prime }}$$
where the $K^{\prime }$ values are power law constants and the $n^{\prime }$ values are frequency exponents.

### 3.9. Determination of Gel Properties of Soy Protein Emulsion

#### 3.9.1. Textural Properties

According to the method of Bourne et al. [58], the texture of soy protein emulsion gel was analyzed. The hardness, viscosity, elasticity, and cohesiveness of the emulsion gel were measured by using a texture analyzer (TA-XT plus, Stable Micro System, Godalming, UK). The test conditions were p/36 probe, 1 mm/s before test, 5 mm/s after test, 5.0 g trigger force, 30% deformation of gel, and 5 s interval between two compression cycles.

#### 3.9.2. Water-Holding Capacity

The water-holding capacity of soy protein emulsion gel was determined according to the method of Guo et al. [15] with slight modification. The polyethylene (PE) pipe filled with 5 mL of emulsion gel was centrifuged in a high-speed centrifuge (GL-20G-Ⅱ, Anting, Shanghai, China) (10,000 r/min, 15 min, 4 °C) to remove the water after centrifugation. The water-holding capacity is calculated as follows:(4)$$\mathrm{WHC}(\%)=\frac{w_{r}-w_{0}}{w_{t}-w_{0}}\times 100\%$$
where $w_{t}$ is the weight of the precentrifugal emulsion gel—g; $w_{r}$ is the weight of emulsion gel after centrifugation to remove moisture—g; and $w_{0}$ is the weight of PE pipe—g.

#### 3.9.3. Hydration Characteristics

The hydration characteristics of soy protein emulsion gel samples were determined by using a low-field nuclear magnetic resonance analyzer (MicroMR02–025 V, Niumag, Suzhou, China) based on Zhang et al. [31]. A 5 mL emulsion was added to GDL, stirred for 30 s, and mixed well. After mixing, the sample was placed in a chromatographic bottle and heated in a 40 °C water bath for 2 h. Then, the soy protein emulsion gel was put into a 4 °C refrigerator overnight, and the emulsion gel was kept at room temperature for 30 min before the test. The test conditions were as follows: proton resonance frequency 22 MHz and measuring temperature 32 °C. Test Parameters: repeat scan 8 times, repeat interval TR 6500 ms; sampling interval 100 s, ECHO number 12,000.

### 3.10. Microstructure of Emulsion Gel

Scanning Electron Microscopy (SEM)

Based on the method of Mantovani [59], the emulsion gel samples were observed by using a SU8010 field emission Scanning electron microscope. The emulsion gel was cut into 3 × 3 × 1 mm pieces and fixed in glutaraldehyde solution. The next day, it was rinsed three times with PBS (0.01 mol/L, pH 6.8) and fixed with concentrated sulfuric acid. Then, 50%, 70%, and 90% ethanol were dehydrated once, and the anhydrous ethanol was dehydrated three times. The mixture of glutaraldehyde and ethanol = 1:1 (*v*/*v*) was replaced for 15 min, and then the mixture was put into pure glutaraldehyde solution for overnight freezing and lyophilized. The freeze-dried samples were then sprayed with gold and placed in the scanning electron microscope for observation.

According to the method of Xi [59], the microstructure of the emulsion gel was observed by a TCS SP8 laser confocal microscope. Nile red (0.1%, *w*/*v*) and Nile blue (1%, *w*/*v*) were dissolved in isopropanol to prepare the dye solution under light-proof conditions. Nile Red was the oil phase dye and Nile blue was the protein dye. An amount of 50 μL of each dye was mixed with 1 mL of the new emulsion, and the emulsion was dyed at 25 °C for 30 min by magnetic stirring. GDL (final pH 4.5) was added to the dyed emulsion, stirred for 30 s in the dark, 1 μL of the mixed emulsion was immediately placed on the slide, the slide was covered, and the bubbles were removed. Under light-proof conditions and 40 °C for 2 h, the emulsion gel was observed under a confocal microscope. The excitation wavelength was 633 nm and 488 nm, respectively.

### 3.11. Statistical Analysis

All the tests were performed in triplicate. The results are shown as the mean ± standard deviation. The data were analyzed with SPSS 20.0 software, and *p* > 0.05 was considered to indicate a significant difference. Data processing and chart drawing were carried out by using Origin 8.5 and other software.

## 4. Conclusions

Microwave modification, alkali polyphenol covalent binding, and microwave-alkali polyphenol covalent binding can improve the gel properties of soy protein emulsion gels. After different modification methods, the average particle size of the soy protein emulsion decreased, the zeta potential increased, the interfacial protein content increased, the apparent viscosity decreased, and a more stable emulsion was formed. After adding GDL to the soy protein emulsion, the texture properties and water-holding capacity of the modified soy protein emulsion gel increased. The results of rheological analysis showed that emulsion gels with solid properties were formed in different emulsions, and the modified emulsions had a stronger network structure. Low-field results showed that the modified emulsion gel had a stronger ability to bind water molecules and a higher content of water in the gel network. The SEM and CLSM results showed that the emulsion gel had a porous structure, the microstructure was denser and more homogeneous after modification, and the oil droplets were better embedded in the network structure. The results showed that the better the properties of the soy protein emulsion were, the better the properties of the soy protein emulsion gel. At the same time, compared with the single modification method, the composite modification had the most significant effect on the properties of soy protein emulsion and emulsion gel. This study lays a theoretical foundation for the application of modified soy protein in a protein emulsion gel system.

## Figures and Tables

**Figure 1 molecules-27-03458-f001:**
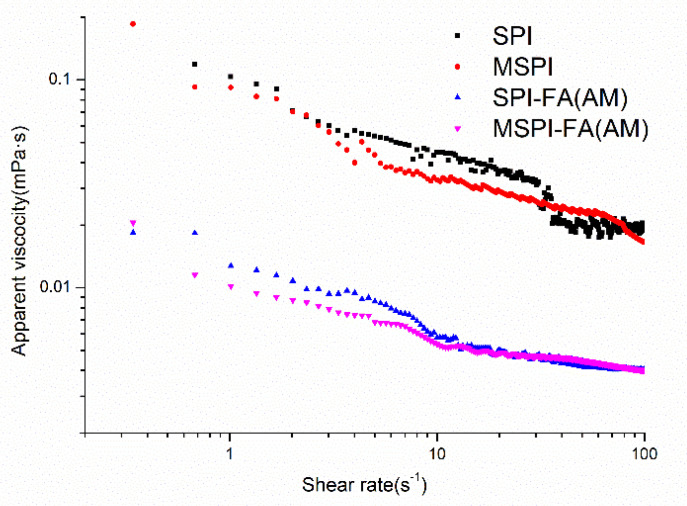
Rheology of emulsions: Apparent viscosity-shear rate flow curve of the emulsion, as the increased from 0.01 to 100 s^−1^.

**Figure 2 molecules-27-03458-f002:**
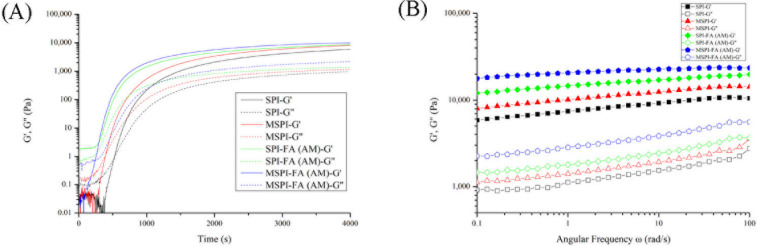
Time sweep (**A**) and frequency sweep (**B**) of emulsion gels induced by different modified soybean protein isolates.

**Figure 3 molecules-27-03458-f003:**
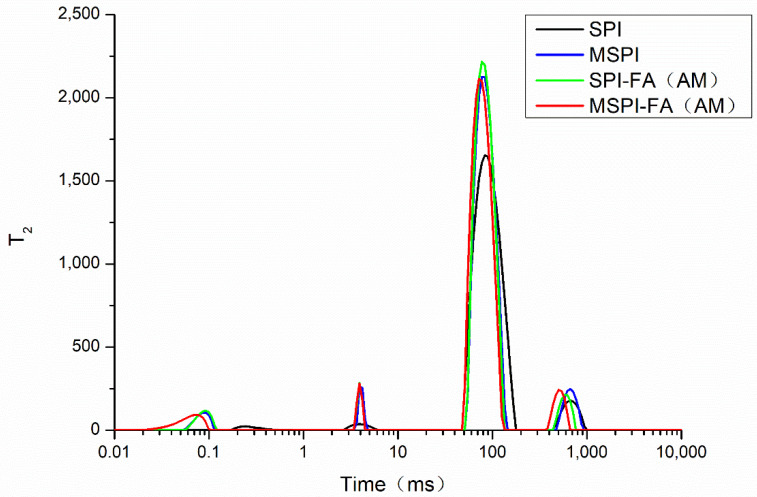
The spin–spin relaxation time (T_2_) induced by different modified soybean protein isolates.

**Figure 4 molecules-27-03458-f004:**
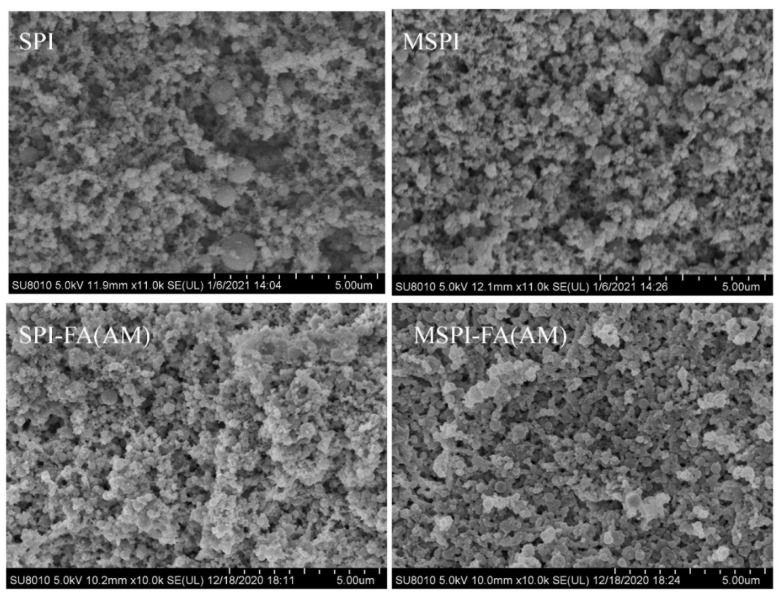
SEM microscopy of emulsion gels induced by different modified soybean protein isolates.

**Figure 5 molecules-27-03458-f005:**
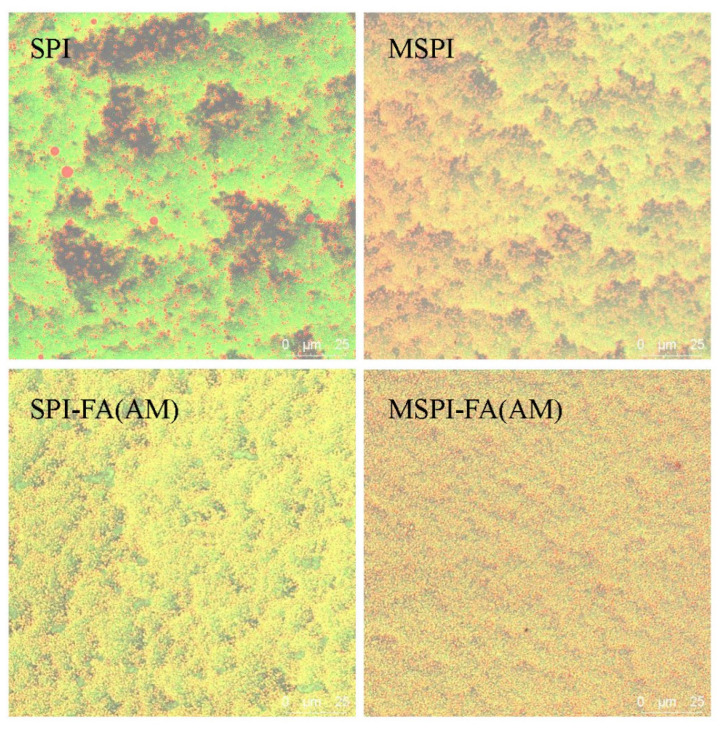
CLSM microscopy of emulsion gels induced by different modified soybean protein isolates.

**Figure 6 molecules-27-03458-f006:**
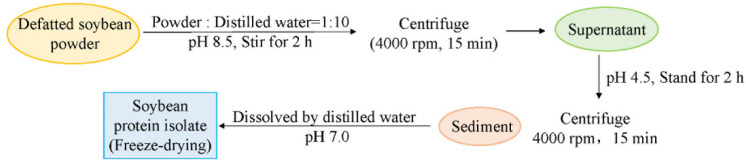
Specific extraction process for SPI.

**Table 1 molecules-27-03458-t001:** Characteristics (mean particle size, **polydispersity index**, zeta-potential, and AP) of emulsions stabilized by different modified soybean protein isolates.

	Mean Particle Size (nm)	PDI	Zeta-Potential (mV)	AP (%)
SPI	282.77 ± 0.67 ^d^	0.281 ± 0.007 ^c^	−26.20 ± 0.30 ^a^	40.27 ± 0.65 ^a^
MSPI	261.63 ± 6.32 ^c^	0.241 ± 0.001 ^b^	−28.17 ± 0.42 ^b^	55.50 ± 1.13 ^b^
SPI-FA (AM)	243.07 ± 0.25 ^b^	0.222 ± 0.012 ^ab^	−29.60 ± 0.10 ^c^	69.67 ± 2.52 ^c^
MSPI-FA (AM)	230.83 ± 1.79 ^a^	0.209 ± 0.017 ^a^	−30.40 ± 0.10 ^d^	72.88 ± 0.74 ^d^

^a–d^ values with different superscript letters in each column are significantly different (*p* < 0.05).

**Table 2 molecules-27-03458-t002:** The apparent viscosity (η), consistency index (K), and flow behavior index (*n*) of emulsions stabilized by different modified soybean protein isolates.

	η_∞_ (mPa·s)	K	*n*	R^2^
SPI	0.019 ± 0.001 ^c^	0.055 ± 0.004 ^d^	0.616 ± 0.010 ^a^	0.940 ± 0.003
MSPI	0.015 ± 0.001 ^c^	0.026 ± 0.003 ^c^	0.754 ± 0.004 ^b^	0.912 ± 0.010
SPI-FA (AM)	0.004 ± 0.001 ^b^	0.018 ± 0.002 ^b^	0.768 ± 0.004 ^c^	0.930 ± 0.013
MSPI-FA (AM)	0.003 ± 0.001 ^a^	0.009 ± 0.001 ^a^	0.789 ± 0.004 ^d^	0.960 ± 0.004

^a–d^ values with different superscript letters in each column are significantly different (*p* < 0.05).

**Table 3 molecules-27-03458-t003:** Crossover points between G′ and G″ of different emulsion gels.

	t_c_	G_c_
SPI	514.954	24.3002085
MSPI	393.585	26.5423884
SPI-FA (AM)	--	--
MSPI-FA (AM)	237.538	21.65209879

**Table 4 molecules-27-03458-t004:** Frequency dependence parameters of different emulsion gels analyzed.

	G′ = K′·ω ^n′^		
	K′	n′	R^2^
SPI	7427.1 ± 18.4 ^a^	0.095 ± 0.003 ^d^	0.990 ± 0.003
MSPI	10,131.8 ± 28.2 ^b^	0.086 ± 0.002 ^c^	0.993 ± 0.001
SPI-FA (AM)	14,568.1 ± 37.3 ^c^	0.071 ± 0.001 ^b^	0.998 ± 0.001
MSPI-FA (AM)	17,386.7 ± 11.8 ^d^	0.048 ± 0.001 ^a^	0.996 ± 0.003

K′ values are power law constants, n′ values are considered as frequency exponents, and ω is the angular frequency. ^a–d^ values with different superscript letters in each column are significantly different (*p* < 0.05).

**Table 5 molecules-27-03458-t005:** Texture properties and WHC of emulsion gels induced by different modified soybean proteins.

	Hardness (g)	Springiness (mm)	Cohesiveness	Chewiness	WHC (%)
SPI	91.44 ± 0.17 ^a^	0.827 ± 0.003 ^a^	0.415 ± 0.004 ^a^	31.38 ± 0.19 ^a^	76.9 ± 0.2 ^a^
MSPI	109.46 ± 0.20 ^b^	0.931 ± 0.006 ^b^	0.534 ± 0.005 ^b^	54.44 ± 0.25 ^b^	79.3 ± 0.1 ^b^
SPI-FA (AM)	226.30 ± 5.02 ^c^	0.961 ± 0.019 ^c^	0.572 ± 0.001 ^c^	124.37 ± 5.37 ^c^	97.6 ± 0.5 ^c^
MSPI-FA (AM)	264.91 ± 4.67 ^d^	0.989 ± 0.001 ^d^	0.606 ± 0.007 ^d^	158.75 ± 1.10 ^d^	99.0 ± 0.3 ^d^

^a–d^ values with different superscript letters in each column are significantly different (*p* < 0.05).

**Table 6 molecules-27-03458-t006:** Water state of emulsion gels induced by different modified soybean protein isolates.

	T_22_ (ms)	T_23_ (ms)	P_22_ (%)	P_23_ (%)
SPI	83.30 ± 0.20 ^d^	694.66 ± 6.81 ^d^	84.83 ± 0.09 ^a^	12.74 ± 0.12 ^d^
MSPI	77.55 ± 0.03 ^c^	666.99 ± 0.00 ^c^	87.68 ± 0.63 ^b^	6.98 ± 0.10 ^c^
SPI-FA (AM)	58.17 ± 0.48 ^b^	439.76 ± 0.00 ^b^	89.54 ± 0.02 ^c^	4.38 ± 0.13 ^b^
MSPI-FA (AM)	47.69 ± 0.00 ^a^	415.10 ± 5.03 ^a^	90.94 ± 0.04 ^d^	2.55 ± 0.12 ^a^

^a–d^ values with different superscript letters in each column are significantly different (*p* < 0.05).

## Data Availability

Not applicable.

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
