# Peer review of "Gel Property of Soy Protein Emulsion Gel: Impact of Combined Microwave Pretreatment and Covalent Binding of Polyphenols by Alkaline Method"

_molecules, 2022, doi:10.3390/molecules27113458_

Round 1
Reviewer 1 Report
Minor spelling mistakes, fig 3
Line 370, ferulic word from ferulic acid is underlined, why?
Many abbreviations are not explicit (PBS, FA - (ferulic acid?), PE pipe)
Missing diagram for LF-NMR; The data in the table is not relevant. What was considered a standard?
What does "henan digram" mean?
Fig 2 illegible (conditions for performing SEM)
Author Response
Thank you very much for your devoted time to reading and commenting on our manuscript. We appreciate your rigorous and conscientious effort for the improvement of the quality of the manuscript. Moreover, we are grateful for the opportunity to respond to the editor and reviewer on their relevant comments.
We have modified the manuscript accordingly and each point that was raised is clarified below. All changes were also highlighted (red) throughout the manuscript.
The manuscript has been resubmitted and we look forward to your positive response.
- Minor spelling mistakes, fig 3
Authors: Thanks to you for your comment. We accept this suggestion and have modified the word “powder”.(Line 380)
- Line 370, ferulic word from ferulic acid is underlined, why?
Authors: We accept this suggestion and have removed the underline.(Line 383)
- Many abbreviations are not explicit (PBS, FA - (ferulic acid?), PE pipe)
Authors: We accept this suggestion .The full name is added before each abbreviation when it first appears. Such as “ferulic acid(FA)”, “phosphate buffer saline solution (PBS)”, “polydispersity index and (PDI)”, “polyethylene (PE) pipe”.(Line 149, 385-386, 432, 493)
- Missing diagram for LF-NMR; The data in the table is not relevant. What was considered a standard?
Authors: We accept this suggestion. Diagram for LF-NMR have been added. (Line 335)
The spin-spin relaxation time (T2) is the time corresponding to each peak in the relaxation diagram. The shorter the peak time (that is, the smaller the T2) , the more difficult the water molecules are to flow. T2 contains T2b, T21, T22, T23. In general, T2B (< 1 ms) represents strong-binding water that is electrostatic and hydrogen-bonded to protein molecules, while T21(1~10 ms) represents weak-binding water that is tightly bound to protein molecules T22(10 ~ 100 ms) represents water that is not easy to flow in the pores of the gel network, and T23(>100 ms) represents free-flowing water outside the gel network. P2B, P21, P22, P23 represent the peak area proportion (%) under T2b, T21, T22, T23 time, respectively, which shows the percentage content of strong-binding water, weak-binding water, hard-flowing water and free-flowing water. T2B and T21 are not discussed in this test because the binding water is not normally affected.
Table 5 shows T22, T23, P22, and P23 of soy protein emulsion gels. The results showed that the relaxation time of T22 and T23 decreased (p < 0.05) , the content of flowing water increased (p < 0.05) , and the content of free flowing water decreased (p < 0.05) , this is related to the expansion of the structure of microwave modified soy protein, the decrease of the average particle size of the emulsion, the formation of cross-linking units per volume, the enhancement of the gel structure, the increase of the hard-flowing water content in the emulsion and the decrease of the free-flowing water content. Relaxation time T22 and T23 decreased significantly (p < 0.05) , P22 content increased significantly (p < 0.05) , P23 content decreased significantly (p < 0.05) , flow water content did not increase easily, this may be due to the covalent binding of proteins with polystyrene to form a dense, uniform emulsion network, resulting in more water trapped in the gel in the form of less flowing water. T22 and T23 decreased further (p < 0.05) , P22 increased further (p < 0.05) , P23 decreased further (p < 0.05) . The results show that microwave-alkali-phenol covalent modification can improve the binding ability of water molecules in porous structure of gel and increase the content of water molecules. The results showed that microwave modification and covalent modification had synergistic effect on alkaline polyphenol. The experimental results are consistent with the water holding capacity of the emulsion gel.
- What does "henan digram" mean?
Authors: We accept this suggestion and “henan” have been changed to “relaxation”.(Lines 305-306)
- Fig 2 illegible (conditions for performing SEM)
Authors: We accept this suggestion. The figures have been modified and the SEM conditions have been supplemented.(Lines 511-529)
Reviewer 2 Report
This manuscript deals with the influence of microwave treatment on soy protein emulsions which were previously modified or not by covalent binding of ferulic acid (polyphenol) with them in alkaline media. The research is properly designed, and a sufficient number of techniques has been used in this study. However, only one figure is provided in results and discussion section. For example, viscosity curves mentioned in section 2.1.3 as well as texture profiles obtained from TA-TXT instrument as describe in section 3.9. should be displayed in the manuscript. The values of parameters presented in Tables 2 and 4 is not the whole information that can be obtained from the graphs required. Similarly, the RMN plot could be included in the manuscript.
Lines 116-119: Reference cited (21) is not related to soybean protein but α-lactalbumin. Additionally, no reference is provided to support the last sentence of this paragraph. In fact, none of the references (18-25) cited in section 2.1.1 are referred to soybean protein. More appropriate references should be provided.
Table 1: What do the authors mean with PDI? This acronym is not defined in the previous text neither the method used to obtain these values is described in its respective section. Is it Polydispersity Index?
Lines 154-155: The same sentence that was stated in lines 118-119 is repeated. In this case, reference 27 is related to soybean protein. Please, revise the references are properly cited in both subsections.
Section 2.1.3: This section should include the viscosity curves of the emulsions under study. In fact, the first sentences cannot be mentioned using reference 21 (perhaps 27) instead of displaying theses graphs.
Table 2: What are the values of the apparent viscosity presented in this table? Apparent viscosity is a function of the shear rate, and an unique value is presented in the this table.
Figure 1B: The Y-axis can be reduced in one order of magnitude as the legend is reconditioned.
Line 206: Elastic modulus (G’) should be written in capital letter.
Lines 251-255: Have the authors considered that microwave treatment could provide energy enough to exposure buried groups via protein denaturation like heat treatments?
Lines 261-293: References 41, 33, 14, 45 are not directly related with which is stated in their respective previous sentences. In fact, most of the references provided are related to ultrasonic instead to microwave treatments.
Lines 309-321: This part of the text could be rewritten in order to be clearer. Some ideas seem to repeat alongside these sentences.
Lines 299, 310, 316 and 321: The parameter T2 mentioned in these lines is not showed in Table 5 and neither was displayed the henan diagram mentioned in line 299.
Line 365: Missing word (isolate) before SPI acronym.
Figure 3. Why a pH value of 8.5 is used to extract soybean protein from the deffated soybean powder and pH7 to precipitate the protein previously solubilized? Can any reference or solubility curve be provided? Revise English spelling in this figure. Expess properly the rotational speed (rpm).
Line 375: Change NaOH instead of NaoH.
Lines 371, 380, 390, 399, 415, 426, 436, 465, 474 and 483. Different references are cited in these lines referred to the methods used in this study. However, they are not standard or initial methods described in the scientific literature to carry out those tests. In fact, the references cited mention other authors to refer to these methods. So, initial or standard methods should be cited. In addition, rheological methods described in subsections 3.7.3 and 3.8 are the procedure to carry out these tests but not a particular method defined by the cited authors.
Line 430 and equation 2: Change γ for
Lines 439-442: At what temperature was set to carry out the strain scan?
Lines 439, 443 and 451: Use the word ‘sweep’ instead ‘scan’
Author Response
Thank you very much for your devoted time to reading and commenting on our manuscript. We appreciate your rigorous and conscientious effort for the improvement of the quality of the manuscript. Moreover, we are grateful for the opportunity to respond to the editor and reviewer on their relevant comments.
We have modified the manuscript accordingly and each point that was raised is clarified below. All changes were also highlighted (red) throughout the manuscript.
The manuscript has been resubmitted and we look forward to your positive response.
- This manuscript deals with the influence of microwave treatment on soy protein emulsions which were previously modified or not by covalent binding of ferulic acid (polyphenol) with them in alkaline media. The research is properly designed, and a sufficient number of techniques has been used in this study. However, only one figure is provided in results and discussion section. For example, viscosity curves mentioned in section 2.1.3 as well as texture profiles obtained from TA-TXT instrument as describe in section 3.9. should be displayed in the manuscript. The values of parameters presented in Tables 2 and 4 is not the whole information that can be obtained from the graphs required. Similarly, the RMN plot could be included in the manuscript.
Authors: Thanks to you for your comment. We accept this suggestion and have modified the manuscript accordingly and each point raised by reviewer #2. All changes were also highlighted (red) throughout the manuscript.
- Lines 116-119: Reference cited (21) is not related to soybean protein but α-lactalbumin. Additionally, no reference is provided to support the last sentence of this paragraph. In fact, none of the references (18-25) cited in section 2.1.1 are referred to soybean protein. More appropriate references should be provided.
Authors: We accept this suggestion and the and references’ number have been revised in our manuscript.(Lines 116-119)
- Table 1: What do the authors mean with PDI? This acronym is not defined in the previous text neither the method used to obtain these values is described in its respective section. Is it Polydispersity Index?
Authors: We accept this suggestion and the and an explanation(“polydispersity index”) has been added to refer to the polydispersity index. The retrieval method has been added in the material and method in our manuscript.(Lines 136, 412-422)
- Lines 154-155: The same sentence that was stated in lines 118-119 is repeated. In this case, reference 27 is related to soybean protein. Please, revise the references are properly cited in both subsections.
Authors: We accept this suggestion and the references and references’ number have been revised in our manuscript.(Lines 155-156)
- Section 2.1.3: This section should include the viscosity curves of the emulsions under study. In fact, the first sentences cannot be mentioned using reference 21 (perhaps 27) instead of displaying theses graphs.
Authors: We accept this suggestion and the viscosity curves of the emulsions have been added in our manuscript. The reference’s number has been revised.(Lines 189, 164-188)
- Table 2: What are the values of the apparent viscosity presented in this table? Apparent viscosity is a function of the shear rate, and an unique value is presented in the this table.
Authors: We accept this suggestion. The apparent viscosity values are in the first column of the table 2.(Lines 192-194)
- Figure 1B: The Y-axis can be reduced in one order of magnitude as the legend is reconditioned.
Authors: We accept this suggestion . The legend has been reconditioned and the Y-axis has been reduced in one order of magnitude in our manuscript.(Line 247)
- Line 206: Elastic modulus (G’) should be written in capital letter.
Authors: We accept this suggestion and Elastic modulus (G’) has been written in capital letter.(Line 210)
- Lines 251-255: Have the authors considered that microwave treatment could provide energy enough to exposure buried groups via protein denaturation like heat treatments?
Authors: We accept this suggestion. The authors suggest that microwave processing could provide sufficient energy to expose buried groups through denaturation.(Lines 257-261)
- Lines 261-293: References 41, 33, 14, 45 are not directly related with which is stated in their respective previous sentences. In fact, most of the references provided are related to ultrasonic instead to microwave treatments.
Authors: We accept this suggestion and the and references’ number have been revised in our manuscript. (Lines 268-299)
- Lines 309-321: This part of the text could be rewritten in order to be clearer. Some ideas seem to repeat alongside these sentences.
Authors: We accept this suggestion. We’ve modified this section. In this part, the meanings of T2(T2b, T21, T22 and T23) and P2B, P21, P22, P23 are introduced firstly; and then the hydration properties of microwave modified protein emulsion gels are introduced; finally, the hydration characteristics of the gel of the composite modified protein emulsion were introduced.(Lines 316-328)
- Lines 299, 310, 316 and 321: The parameter T2 mentioned in these lines is not showed in Table 5 and neither was displayed the henan diagram mentioned in line 299.
Authors: We accept this suggestion and the relaxation diagram has been added. T2, which includes T2b, T21, T22, and T23, has been added in our manuscript. As noted in our manuscript, T2b, T21, T22 and T23respectively shows the percentage of strong-binding water, weak-binding water, hard-flowing water and free-flowing water. T2B and T21 are not discussed in this text because the binding water is not normally affected. hard-flowing water and free-flowing water. T2B and T21 are not discussed in this text because the binding water is not normally affected.(Lines 317, 323, 328)
- Line 365: Missing word (isolate) before SPI acronym.
Author: We accept this suggestion. “isolation” has been added between “protein” and “(SPI)” in our manuscript.(Line 376)
- Figure 3. Why a pH value of 8.5 is used to extract soybean protein from the deffated soybean powder and pH7 to precipitate the protein previously solubilized? Can any reference or solubility curve be provided? Revise English spelling in this figure. Expess properly the rotational speed (rpm).
Author: We accept this suggestion. The method of protein extraction was based on the reference literature. The soy protein was extracted at pH 8.5, and the soy protein was precipitated at pH 4.5. Finally, the precipitate was dissolved at pH 7. English spelling has been revised(powder). We have changed the rotational speed to “rpm”.(Line 380)
- Line 375: Change NaOH instead of NaoH.
Author: We accept this suggestion . We have changed NaOH instead of NaoH.(Line 388)
- Lines 371, 380, 390, 399, 415, 426, 436, 465, 474 and 483. Different references are cited in these lines referred to the methods used in this study. However, they are not standard or initial methods described in the scientific literature to carry out those tests. In fact, the references cited mention other authors to refer to these methods. So, initial or standard methods should be cited. In addition, rheological methods described in subsections 3.7.3 and 3.8 are the procedure to carry out these tests but not a particular method defined by the cited authors.
Author: We accept this suggestion . Some of the research methods used in this paper are slightly modified from the standard methods. The methods used in the 371, 380, 390, 399, 415, 426, 436, 465, 474 and 483 lines of reference are the best fit for our study, so we chose to cite these documents. The methods of conducting these experiments are described in sections 3.7.3 and 3.8.
- Line 430 and equation 2: Change γ for
Author: We accept this suggestion. We’ve changed the format of γ.(Lines 455-457)
- Lines 439-442: At what temperature was set to carry out the strain scan?
Author: We accept this suggestion. “at 40 °C” was added in our manuscript. The strain scan carried out at 40 °C. Because the gel was formed at 40 °C.(Line 469)
- Lines 439, 443 and 451: Use the word ‘sweep’ instead ‘scan’
Author: We accept this suggestion. We’ve replaced “Scan”with “Sweep”.(Lines 467, 471 and 479)
Round 2
Reviewer 2 Report
In general, authors have accepted the suggestions made in the previous revision. However, as new figures were proposed to be included some new considerations are listed below.
- Figure 1. A log-log scale graphic should be used to display viscosity curves. Additionally, write -1 as superscript in this figure caption.
- According to data displayed in figure 1, the values of apparent viscosity showed in Table 2 should be the so called ‘limiting high-shear viscosity’, η∞, which represents the viscosity at shear rates approaching an infinitely high value. Please, modified this symbol in Table 2 and name specifically this characteristic viscosity in the text in section 2.1.3.
- Line 200. Change ‘time-scan results’ by ‘time-sweep tests’.
- Line 202. What means ‘diagonal frequencies’? ¿Is this sentence correct?
- Crossover point between G’ and G” in Figure 2A could be useful to discuss the anticipation of the gelation. At this crossover point you would have two parameters, Gc and tc. The first one would be related to the strength of gel and the second one could be useful to discuss how the use of microwave and polyphenols would anticipate the gelation process. This parameters could be showed in a table, a graph, or presented throughout the text.
- It has not been showed any graphic of texture profile. At least one graphic and the procedure to obtain the different parameters presented in Table 4 should be presented.
- Line 376. ‘Isolation’ is the process, the correct word should be ‘isolate’ (Soy protein isolate).
- Line 415. The average particle… This sentence seems to be incompleted. In addition, the whole paragraph (lines 413-422) are very similar to the other one existing in lines 433-438.
- Line 463. Write ‘serrated’ instead ‘roughish’.
Author Response
Reviewer
- Figure 1. A log-log scale graphic should be used to display viscosity curves. Additionally, write -1 as superscript in this figure caption.
Authors: Thanks to you for your comment. We accept this suggestion. We have modified the title of figure and figure. Lines 190-192.
- According to data displayed in figure 1, the values of apparent viscosity showed in Table 2 should be the so called ‘limiting high-shear viscosity’, η∞, which represents the viscosity at shear rates approaching an infinitely high value. Please, modified this symbol in Table 2 and name specifically this characteristic viscosity in the text in section 2.1.3.
Authors: Thanks to you for your comment. We accept this suggestion. We have changed the symbols and added an explanation to the manuscript. Lines 168-169, 194-195.
- Line 200. Change ‘time-scan results’ by ‘time-sweep tests’.
Authors: Thanks to you for your comment. We accept this suggestion and we have changed ‘time-scan results’ by ‘time-sweep tests’. Line 201.
- Line 202. What means ‘diagonal frequencies’? ¿Is this sentence correct?
Authors: Thanks to you for your comment. We accept this suggestion. We have changed “diagonal frequencies” by “angular frequencies” and revised the sentence. Line 205.
- Crossover point between G’ and G” in Figure 2A could be useful to discuss the anticipation of the gelation. At this crossover point you would have two parameters, Gc and tc. The first one would be related to the strength of gel and the second one could be useful to discuss how the use of microwave and polyphenols would anticipate the gelation process. This parameters could be showed in a table, a graph, or presented throughout the text.
Authors: Thanks to you for your comment. We accept this suggestion. We have added tables and explanations of Gc and tc and discussed the rate of gel formation in the manuscript. Lines 254, 201-204, 217, 219.
- It has not been showed any graphic of texture profile. At least one graphic and the procedure to obtain the different parameters presented in Table 4 should be presented.
Authors: Thanks to you for your comment. We accept this suggestion. There are many indexes of texture measurement, each of which provides tables and graphs, but there are many tables and graphs in articles. In the meantime, we refer to the latest literature and they provide only data in the section dealing with gel texture. For example:
- Properties of soybean protein isolate/curdlan based emulsion gel for fat analogue: Comparison with pork backfat. International Journal of Biological Macromolecules, 2022,206:481–488.
â‘¡ Differences in the physicochemical, digestion and microstructural characteristics of soy protein gel acidified with lactic acid bacteria, glucono-δ-lactone and organic acid. International Journal of Biological Macromolecules, 2021,185:462–470
â‘¢ Emulsion gels stabilized by soybean protein isolate and pectin: Effects of high intensity ultrasound on the gel properties, stability and β-carotene digestive characteristics. Ultrasonics Sonochemistry, 2021,79:105756
â‘£ Physicochemical, rheological and digestive characteristics of soy protein isolate gel induced by lactic acid bacteria. Journal of Food Engineering, 2021,292:110243
We use the same equipment as â‘ and â‘¡. All of them are TA-XT plus, Stable Micro Systems Ltd. . , UK.
- Line 376. ‘Isolation’ is the process, the correct word should be ‘isolate’ (Soy protein isolate).
Authors: Thanks to you for your comment. We accept this suggestion and we have change “isolation” by “isolate”. Lines 47, 381.
- Line 415. The average particle… This sentence seems to be incompleted. In addition, the whole paragraph (lines 413-422) are very similar to the other one existing in lines 433-438.
Authors: Thanks to you for your comment. We accept this suggestion. This section overlaps with the others and we have revised and placed the manuscript in the correct place. Line 426.
- Line 463. Write ‘serrated’ instead ‘roughish’.
Authors: Thanks to you for your comment. We accept this suggestion and we have changed “roughish” by “serrated”. Line 457.
